# Clinical Utility of Plasma Cell-Free DNA EGFR Mutation Analysis in Treatment-Naïve Stage IV Non-Small Cell Lung Cancer Patients

**DOI:** 10.3390/jcm11041144

**Published:** 2022-02-21

**Authors:** Bo-Guen Kim, Ja-Hyun Jang, Jong-Won Kim, Sun Hye Shin, Byeong-Ho Jeong, Kyungjong Lee, Hojoong Kim, O Jung Kwon, Myung-Ju Ahn, Sang-Won Um

**Affiliations:** 1Division of Pulmonary and Critical Care Medicine, Department of Medicine, Samsung Medical Center, Sungkyunkwan University School of Medicine, Seoul 06351, Korea; kbg1q2w3e@gmail.com (B.-G.K.); freshsunhye@gmail.com (S.H.S.); myacousticlung@gmail.com (B.-H.J.); kj2011.lee@gmail.com (K.L.); hjk3425@skku.edu (H.K.); ojkwon@skku.edu (O.J.K.); 2Department of Laboratory Medicine and Genetics, Samsung Medical Center, Sungkyunkwan University School of Medicine, Seoul 06351, Korea; jahyun.jang@gmail.com (J.-H.J.); kimjw@skku.edu (J.-W.K.); 3Division of Hematology-Oncology, Department of Medicine, Samsung Medical Center, Sungkyunkwan University School of Medicine, Seoul 06351, Korea; silkahn@skku.edu; 4Department of Health Sciences and Technology, Samsung Advanced Institute for Health Sciences & Technology, Sungkyunkwan University, Seoul 06355, Korea

**Keywords:** plasma cell-free DNA, EGFR mutation, carcinoembryonic antigen, non-small cell lung cancer, EGFR-TKI

## Abstract

Background: Plasma cell-free Deoxyribo nucleic acid epidermal growth factor receptor (EGFR) mutation tests are widely used at initial diagnosis and at progression in stage IV non-small cell lung cancer (NSCLC). We analyzed the factors associated with plasma EGFR mutation detection and the effect of plasma EGFR genotyping on the clinical outcomes of the patients with treatment-naïve stage IV NSCLC. Methods: In this retrospective cohort study, we included subjects with treatment-naïve stage IV NSCLC who underwent plasma EGFR genotyping between 2018 and 2020. The presence of plasma EGFR mutation was determined by real-time polymeric chain reaction. Results: The prevalence of EGFR mutation in this cohort was 52.7% (164/311). Among 164 EGFR mutant subjects, 34 (20.7%) were positive for the plasma EGFR mutation assay only. In multivariable analysis, the detection of plasma EGFR mutation was significantly related to higher serum carcinoembryonic antigen levels, never-smoker status, N3 stage, and brain or intrathoracic metastasis. The time to treatment initiation (TTI) of the plasma EGFR mutation-positive group (14 days) was shorter than that of the plasma EGFR mutation-negative group (21 days, *p* < 0.001). More patients received the 1st line EGFR-TKI in the plasma positive group compared with the tissue positive group. Conclusion: Smoking status and the factors reflecting tumor burden were associated with the detection of plasma EGFR mutation. The plasma EGFR mutation assay can shorten the TTI, and facilitate the 1st line EGFR-TKI therapy for patients with treatment-naïve stage IV NSCLC, especially in the region of high-prevalence of EGFR mutation.

## 1. Introduction

Lung cancer is still the most common cause of cancer-related mortality [1,2]. Recent advances in targeted therapy have significantly improved the treatment outcomes of non-small cell lung cancer (NSCLC). In particular, somatic mutations in epidermal growth factor receptor (EGFR) are important parameters for determining the treatment response to EGFR tyrosine kinase inhibitors (EGFR-TKIs) in NSCLC [3,4]. EGFR mutations are detected in 30–50% of NSCLCs from Asians and 10% of those from Caucasians [5,6,7].

Traditionally, molecular genotyping has been performed using tissues, but with recent technical advances, various liquid biopsy platforms can now be used to detect plasma-circulating tumor deoxyribonucleic acid (DNA) [6,8]. Comprehensive cell-free DNA (cfDNA) analysis has the advantages of a faster turnaround time (TAT) than tissue analysis, less invasiveness, and the ability to obtain genetic information when tissue biopsy is not possible [9,10,11].

The NCCN guidelines and a consensus statement from the International Association for the Study of Lung Cancer (IASLC) also recommend the use of plasma genotyping both at initial diagnosis if sufficient tissue is not available as well as at progression on EGFR-TKIs [12]. However, to the best of our knowledge, few studies have investigated the characteristics of patients who are more likely to be positive in plasma EGFR mutation analysis. Moreover, few studies have been conducted on the clinical utility of plasma EGFR mutation analysis in real-world practice.

Therefore, we investigated factors affecting the positivity of plasma EGFR mutation assay and its effect on the clinical outcomes of patients with treatment-naïve stage IV NSCLC.

## 2. Methods

### 2.1. Study Population and Data Collection

This was a retrospective cohort study conducted at Samsung Medical Center, a referral hospital in South Korea. We included patients with treatment-naïve stage IV adenocarcinoma and NSCLC not otherwise specified, who underwent plasma EGFR mutation assay between January 2018 and December 2020. Patients who underwent plasma EGFR mutation assay during or after chemotherapy, and those who were not finally diagnosed with primary lung cancer, were excluded. Subjects with squamous cell carcinoma, large cell neuroendocrine carcinoma, or pleomorphic carcinoma on final pathology were also excluded.

We gathered the following information from the database: patient-related factors such as age, sex, smoking history, and European Cooperative Oncology Group (ECOG) performance status at diagnosis; cancer-related factors such as the serum carcinoembryonic antigen (CEA) level at diagnosis, histologic type, clinical tumor-node-metastasis (TNM) stage, number of metastatic sites, types of metastatic organs, and EGFR mutation subtypes; biopsy procedure-related factors such as tissue biopsy methods and biopsy sites; and treatment- and treatment response-related factors such as use of chemotherapy, use of EGFR-TKIs, time to treatment initiation (TTI), first treatment response, and death. The CEA levels were divided into three groups using the 1st and 3rd quartiles as the cut-off values. The TTI was defined as the duration of time between the first evaluation of primary lung cancer and initiation of treatment. The tumor was staged using the 8th edition of the American Joint Committee on Cancer TNM staging system [13]. The Response Evaluation Criteria in Solid Tumors (version 1.1, the RECIST Working Group) were used to assess the treatment response [14]. Patient follow-up data were last updated in March 2021.

Patients were divided into four groups according to the EGFR mutation test results from the tissue and plasma EGFR mutation assays: “Tissue (−) & plasma (−),” “Tissue (−) & plasma (+),” “Tissue (+) & plasma (−),” and “Tissue (+) & plasma (+)”.

We obtained approval from the Institutional Review Board (IRB approval no. 2021-01-125) to review and publish information from patient records, and the requirement for informed consent was waived due to the retrospective nature of the study.

### 2.2. Methods of Analysis

About 8.5 mL of whole blood was drawn into a cfDNA collection tube (Roche Molecular Diagnostics, Pleasanton, CA, USA) and stored at room temperature until processing. Plasma was separated by double centrifugation (1600× *g* for 15 min followed by 16,000× *g* for 10 min). If the test did not proceed within 72 h of blood collection, plasma after the first centrifugation was stored at −70 °C. A Cobas cfDNA Sample Preparation Kit (Roche Molecular Diagnostics) was used for cfDNA extraction from 2 mL of doubly centrifuged plasma. The presence of an EGFR mutation was determined by real-time polymeric chain reaction (RT-PCR) using the Cobas EGFR Mutation Test v2 (Roche Molecular Diagnostics), which is approved by the Food and Drug Administration for molecular analysis of liquid biopsy specimens in NSCLC [15].

Tissue biopsy samples were analyzed for EGFR mutations using a peptide nucleic acid clamp kit and RT-PCR [16]. Serum CEA was measured using electrochemiluminescence immunoassay (ECLIA) or immunoradiometric assay (IRMA).

### 2.3. Statistical Analyses

Data are presented as number (%) for categorical variables and median (interquartile range [IQR]) for continuous variables. Data were compared using the chi-squared test or Fisher’s exact test for categorical variables and the Mann–Whitney test for continuous variables. Bonferroni’s method was used for post hoc analysis. The gold standard definition of EGFR mutation-positivity is the detection of an EGFR mutation in either tissue or plasma EGFR mutation assays. If the tissue biopsy could not be performed due to the patient’s condition or technical difficulties and the plasma EGFR assay was positive, the tissue EGFR mutation test was considered false negative. We calculated diagnostic sensitivity, specificity, accuracy, positive predictive value (PPV), and negative predictive value (NPV) [17].

Logistic regression analysis with backward stepwise selection (*p* < 0.05 for entry and *p* > 0.10 for removal of variables) was used to identify factors independently associated with a positive plasma EGFR mutation assay. The Kaplan–Meier method was used to estimate overall survival (OS) after lung cancer diagnosis. All tests were two-sided, and *p* < 0.05 was considered significant. All statistical analyses were performed using SPSS software (ver. 27.0; SPSS Inc., Chicago, IL, USA).

## 3. Results

### 3.1. Study Population

Between January 2018 and December 2020, 311 subjects were included in this study (Figure 1). The median age of the study population was 65 (57–74) years, and 56.6% were female (Table 1). Most of the patients (62.7%) were never smokers. Patients were followed up for a median of 5.7 (3.0–10.4) months after lung cancer diagnosis. N3 disease was present in 56.9% of patients, and M1c in 60.8%. Among the types of metastases, intrathoracic metastasis was the most common (61.7%), followed by bone (44.7%) and brain metastases (30.2%).

The prevalence of EGFR mutation was 52.7% (164/311) in this study. Of the 311 subjects, 147 (47.3%) had two negative tests for EGFR mutation (tissue −/plasma −), 34 (10.9%) had only a positive plasma EGFR mutation test (tissue −/plasma +), and 32 (10.3%) had only a positive tissue EGFR mutation test (tissue +/plasma −). Ninety-eight (31.5%) patients were positive on both tests (tissue +/plasma +) (Table 1). The four subgroups differed in sex, smoking history, CEA level, histology, N stage, M stage, number of metastatic sites, brain metastasis, and EGFR mutation type. In the group with a positive plasma EGFR mutation assay (“tissue −/plasma +” or “tissue +/plasma +”), the number of current smokers was lower, and the number of cases with a CEA level above 94.7 ng/mL was higher. In addition, the proportion of patients with N3 stage, M1c stage, four or more metastatic sites, or brain metastasis was higher.

In 164 patients with positive tissue or plasma EGFR mutation tests, 34 (20.7%) were positive in plasma only. Among 34 subjects with “tissue (−)/plasma (+),” 5 did not undergo tissue biopsy and 28 underwent tissue biopsy but did not have sufficient remaining tissue for the EGFR mutation test. One subject had a negative result from the tissue EGFR assay, while the plasma EGFR mutation assay was positive (Figure 2). Finally, 29 (85.3%) of these 34 patients received EGFR TKI as the 1st line treatment based on plasma EGFR mutation test results (Table 1).

### 3.2. Diagnostic Performance of Tissue and Plasma EGFR Mutation Tests

There was no significant difference between the diagnostic sensitivity of the plasma EGFR mutation assay (80.5% [73.6–86.3%]) and tissue EGFR mutation assay (79.3% [72.3–85.2%]) (*p* = 0.787). There was also no significant difference between the diagnostic accuracy of the plasma EGFR mutation assay (89.7% [85.8–92.9%]) and tissue EGFR mutation assay (89.1% [76.2–85.4%]) (*p* = 0.808). The NPV of the plasma EGFR mutation assay (82.1% [77.1–86.2%]) was also similar to that of the tissue EGFR mutation assay (81.2% [76.2–85.4%]) (*p* = 0.826) (Appendix A).

### 3.3. Factors Associated with Positivity of the Plasma EGFR Mutation Assay

We analyzed the factors affecting plasma EGFR mutation positivity (Table 2). In multivariable analysis, plasma EGFR mutation positivity was significantly associated with never-smoker status (adjusted odds ratio [aOR], 2.83; 95% CI, 1.55–5.20; *p* = 0.001), higher serum CEA levels (>94.7 ng/mL; aOR, 2.98; 95% CI, 1.21–7.35; *p* = 0.018), N3 stage (aOR, 4.22; 95% CI, 1.41–12.62; *p* = 0.010), brain metastasis (aOR, 2.73; 95% CI, 1.39–5.36; *p* = 0.003), and intrathoracic metastasis (aOR, 2.61; 95% CI, 1.38–4.96; *p* = 0.003).

### 3.4. Effect of a Positive Result of the Plasma EGFR Mutation Test on Treatment and Overall Survival

The median TAT of the tissue EGFR mutation assay was 11 (9–13) and 41 (34–49) days according to RT-PCR and next-generation sequencing (NGS), respectively. The median TAT of the plasma EGFR mutation assay was 5 (4–6) days (Table 3). The TTI of the plasma EGFR mutation-positive group (14 days) was shorter than that of the plasma EGFR mutation-negative group (21 days, *p* < 0.001). Among 164 patients with positive tissue or plasma EGFR mutation tests, 154 (93.9%) patients received EGFR-TKIs during the whole treatment period, and 147 (89.6%) patients received EGFR-TKIs as a 1st line treatment. Among 4 groups, the frequency of the 1st line EGFR-TKI therapy was higher in the “tissue (−)/plasma (+)” group (85.3%) compared with the “tissue (+)/plasma (−)” group (78.1%) (Table 1).

Figure 3 shows the survival curves according to positive and negative results of the tissue and plasma EGFR mutation assays. The 2-year OS rates of the plasma EGFR mutation-positive and -negative groups were 92.0% and 76.6%, respectively (*p* = 0.016) (Figure 3A). The 2-year OS rates of the tissue EGFR mutation-positive and -negative groups were 96.1% and 72.9%, respectively (*p* < 0.001) (Figure 3B).

## 4. Discussion

In this cohort of treatment-naïve NSCLC, 20.7% had a positive EGFR result in plasma only, and the treatment decision was based solely on the plasma assay. In multivariable analysis, a higher CEA level, never-smoker status, N3 disease, and presence of brain or intrathoracic metastasis were significantly related to plasma EGFR mutation positivity. The TAT and TTI of the plasma EGFR mutation-positive group were shorter than those of the plasma EGFR mutation-negative group. More patients received the 1st line EGFR-TKI in the plasma positive group compared with the tissue positive group. To the best of our knowledge, this study was the first to report an association between serum CEA levels and a positive plasma EGFR mutation test result.

Clinical application of liquid biopsy using plasma cfDNA has increased rapidly for various solid tumors such as lung cancer, breast cancer, colon cancer, melanoma, stomach, etc. [18,19]. Plasma cfDNA technologies have the potential to identify actionable somatic alterations, tumor mutational burden, mutational signature, and tumor-associated methylation changes [18]. Liquid biopsy using plasma cfDNA can influence the early detection, monitoring of minimal residual disease, evaluation of early treatment response, and assessment of clonal evolution of various solid tumors [20].

In our study, the Cobas platform was used for plasma EGFR mutation assay, and its diagnostic sensitivity, specificity, and accuracy were 80.5%, 100%, and 89.7%, respectively. In other studies that used the Cobas platform, the sensitivity was about 60–90% and the specificity was 96–100% [21,22,23,24]. In our study, among 38 subjects who were only positive for the plasma EGFR mutation assay, 5 did not undergo tissue biopsy and 28 did not have sufficient tissue specimens for tissue EGFR mutation assay. We confirmed that the plasma EGFR mutation assay could provide important information facilitating treatment decisions in real-world practice. However, due to differences in breadth and sensitivity among cfDNA platforms [23], awareness of the pros and cons of this test is required. Not all tumors shed enough DNA into the peripheral circulation for mutation detection, so clinicians need to recognize the possibility of false-negative results in liquid biopsies. Therefore, it is important to determine the characteristics of patients who are likely to be positive in the EGFR mutation test using liquid biopsy and to benefit from this test.

CEA has been used as a representative tumor marker for NSCLC [25]. In our study, the detection of plasma EGFR mutation was 2.98 times higher when the CEA level was above about 95 ng/mL. Previous studies also demonstrated that the serum CEA level in treatment-naïve NSCLC patients was associated with tissue EGFR mutation [26,27].

Previous studies have demonstrated that the presence of EGFR mutations is associated with distant metastases [28,29]. In addition, previous studies also reported that the presence of EGFR mutation is associated with bone, brain, and liver metastases [28,29,30,31,32]. Previous studies have evaluated the relationship between patient characteristics and the detection of the plasma EGFR mutation [33,34,35,36,37]. One French multicenter study reported that the detection rate of EGFR mutation in plasma tests was about 44 times higher in never-smokers than in ever-smokers [33]. Previous studies also revealed that the detection of plasma EGFR mutation was approximately 3–9 times higher in the patients with higher N stage [34,37], five times higher in the patients with extrathoracic metastasis [34], and 4–10 times higher in the patients with distant organ metastasis such as bone [36,37]. In our study, similar to previous studies, the detection of plasma EGFR mutation was associated with never-smoker and higher tumor burden such as a higher N stage, and brain or intrathoracic metastasis.

In our study, 20.7% of patients obtained a positive result on the plasma mutation assay only. Deng et al. reported that the clinical outcome of patients who initiated EGFR-TKI after liquid biopsy alone was similar to those who underwent tissue biopsy and then initiated treatment [10]. In our study, the 2-year OS rate of the plasma EGFR mutation-positive group (92.0%) was similar to that of the tissue EGFR mutation-positive group (96.1%). The recent IASLC consensus statement also recommends “plasma first approach” if sufficient tissue is not available or concurrent tumor tissue and cfDNA genotyping for treatment-naïve metastatic NSCLC [11].

The median TAT of the plasma EGFR mutation assay (5 days) was shorter than that of the tissue EGFR mutation assay using real-time PCR (11 days) in our study. The shorter TAT ultimately contributed to the shorter TTI of the plasma EGFR mutation-positive group (14 days) compared to that of the plasma EGFR mutation-negative group (21 days). The frequency of EGFR-TKI use as the 1st line treatment was higher in the tissue (−)/plasma (+) group (85.3%) compared with the tissue (+)/plasma (−) group (78.1%). A short TAT of the plasma EGFR mutation test can facilitate the use of EGFR-TKI as the 1st line treatment. Rapid initiation of treatment based on the plasma EGFR mutation assay could improve the clinical outcomes of patients. To the best of our knowledge, this study was the largest to evaluate the clinical utility of plasma EGFR mutation analysis in treatment-naïve stage IV NSCLC, and the first to report an association between serum CEA levels and a positive plasma EGFR mutation test result.

This study had several limitations. First, it was a retrospective cohort study including a Korean population from a single institution. The prevalence of EGFR mutation in this study was 52.7% (164/311). Therefore, the results may not be generalized to groups with a low prevalence of EGFR mutation. Second, our institution started using the plasma EGFR mutation assay in 2018, and the follow-up period for the subjects was relatively short. To evaluate the long-term effect of the plasma EGFR mutation assay, more follow-up data are needed. Third, the serum CEA values were measured using two different tests, ECLIA and IRMA. However, the correlation of the two test methods was fair in previous studies [38,39]. Finally, limitations of plasma RT-PCR assays also need to be considered. Although plasma NGS allows extensive genomic investigations, from targeted gene panels to whole-exome sequencing or whole-genome sequencing [11,40,41], plasma RT-PCR assays have limitations to detect uncommon EGFR mutations and exon 20 insertions. The recent IASLC consensus statement also supports NGS-based approaches rather than non-NGS-based approaches [11]. However, the plasma NGS approach is not feasible in all countries due to its high cost and reimbursement issues. Therefore, concurrent plasma and tissue RT-PCR assays can be useful, especially in Asian populations with a high prevalence of EGFR mutations.

In conclusion, smoking status and the factors reflecting tumor burden were associated with the detection of plasma EGFR mutation. The plasma EGFR mutation assay can overcome the limitation of tumor tissue availability, shorten the TTI, and facilitate 1st line EGFR-TKI therapy for patients with treatment-naïve stage IV NSCLC, especially in the region of high-prevalence of EGFR mutation.

## Figures and Tables

**Figure 1 jcm-11-01144-f001:**
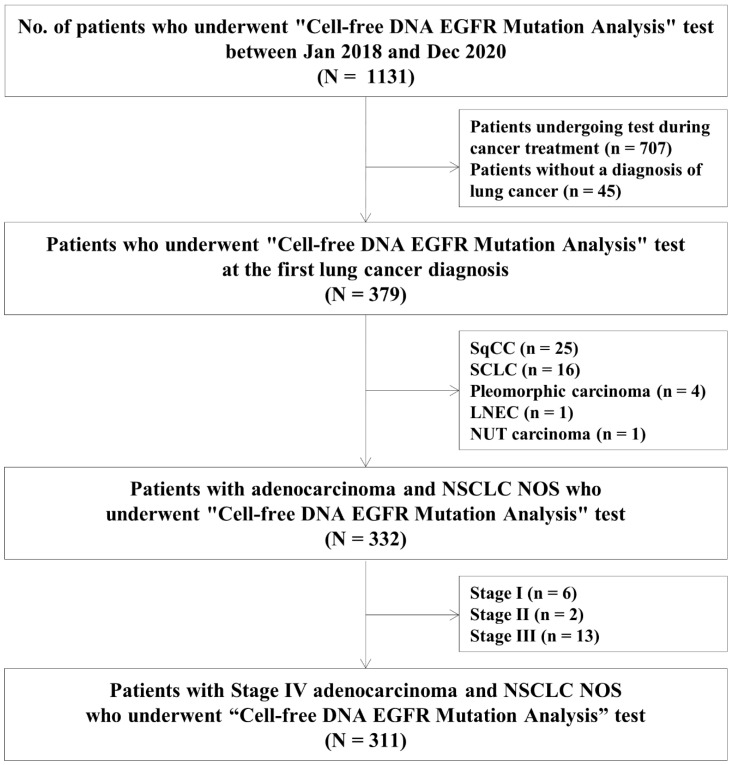
Flow diagram of the full study population. DNA, deoxyribonucleic acid; EGFR, epidermal growth factor receptor; SqCC, squamous cell carcinoma; SCLC, small cell lung cancer; LNEC, large cell neuroendocrine carcinoma; NSCLC NOS, non-small cell lung cancer not otherwise specified.

**Figure 2 jcm-11-01144-f002:**
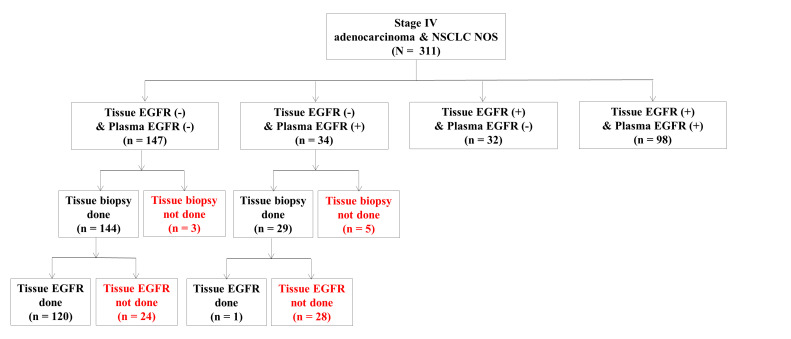
Groups distinguished based on the results of tissue and plasma EGFR mutation assays (positive or negative). Among 34 subjects with “Tissue EGFR (−) & plasma EGFR (+),” there were 5 who did not undergo tissue biopsy and 28 who underwent tissue biopsy but did not have sufficient remaining tissue specimens for EGFR mutation testing. One subject had a negative result from the tissue EGFR assay, while the plasma EGFR mutation assay was positive. NSCLC NOS, non-small cell lung cancer not otherwise specified.

**Figure 3 jcm-11-01144-f003:**
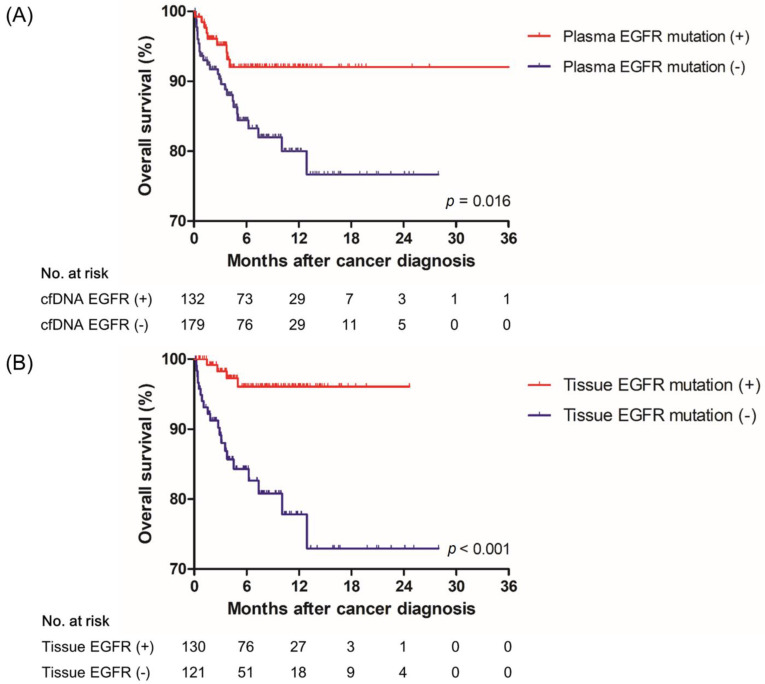
Survival plots of subjects with stage IV NSCLC according to (**A**) plasma and (**B**) tissue EGFR mutation status.

**Table 1 jcm-11-01144-t001:** Baseline characteristics of study subjects according to tissue and plasma EGFR mutation status (N = 311).

Variables	Total (N = 311)	Tissue (−)/ Plasma (−) (N = 147)	Tissue (−)/ Plasma (+) (N = 34)	Tissue (+)/ Plasma (−) (N = 32)	Tissue (+)/ Plasma (+) (N = 98)	*p*
Age, years	65 (57–74)	65 (59–73)	61 (55–79)	66 (54–77)	64 (55–72)	0.732
Sex, female	176 (56.6)	64 (43.5)	19 (55.9)	26 (81.3)	67 (68.4)	<0.001 ^bc^
Smoking history						<0.001 ^bc^
Never smoker	195 (62.7)	71 (48.3)	21 (61.8)	25 (78.1)	78 (79.6)	
Ever smoker	116 (37.3)	76 (51.7)	13 (38.2)	7 (21.9)	20 (20.4)	
ECOG at diagnosis						0.079
ECOG 0–2	297 (95.5)	136 (92.5)	33 (97.1)	31 (96.9)	97 (99.0)	
ECOG 3–4	14 (4.5)	11 (7.5)	1 (2.9)	1 (3.1)	1 (1.0)	
CEA, ng/mL (n = 248)						0.006 ^c^
<3.2 ng/mL	56 (22.6)	39 (33.1)	5 (20.8)	5 (20.0)	7 (8.6)	
3.2–94.7 ng/mL	124 (50.0)	51 (43.2)	13 (54.2)	15 (60.0)	45 (55.6)	
>94.7 ng/mL	68 (27.4)	28 (23.7)	6 (25.0)	5 (20.0)	29 (35.8)	
Histology *						0.022 ^a^
Adenocarcinoma	309 (99.4)	147 (100.0)	32 (94.1)	32 (100.0)	98 (100.0)	
NSCLC NOS	2 (0.6)	0 (0.0)	2 (5.9)	0 (0.0)	0 (0.0)	
Clinical stage at the diagnosis						
T stage						0.705
T1	67 (21.6)	37 (25.2)	8 (23.5)	8 (25.0)	14 (14.3)	
T2	94 (30.2)	38 (25.8)	11 (32.4)	10 (31.3)	35 (35.7)	
T3	57 (18.3)	27 (18.4)	7 (20.6)	5 (15.6)	18 (18.4)	
T4	93 (29.9)	45 (30.6)	8 (23.5)	9 (28.1)	31 (31.6)	
N stage						<0.001 ^bdf^
N0	35 (11.2)	17 (11.6)	2 (5.9)	11 (34.4)	5 (5.1)	
N1	22 (7.1)	14 (9.5)	0 (0.0)	2 (6.3)	6 (6.1)	
N2	77 (24.8)	38 (25.8)	10 (29.4)	11 (34.4)	18 (18.4)	
N3	177 (56.9)	78 (53.1)	22 (64.7)	8 (25.0)	69 (70.4)	
M stage						0.004 ^f^
M1a	91 (29.3)	47 (32.0)	10 (29.4)	16 (50.0)	18 (18.4)	
M1b	31 (10.0)	16 (10.9)	3 (8.8)	5 (15.6)	7 (7.1)	
M1c	189 (60.8)	84 (57.1)	21 (61.8)	11 (34.4)	73 (74.5)	
No. of metastatic sites						<0.001 ^acdf^
1	101 (32.5)	59 (40.1)	9 (26.5)	17 (53.1)	16 (16.3)	
2	73 (23.5)	44 (29.9)	4 (11.8)	9 (28.1)	16 (16.3)	
3	37 (11.9)	23 (15.6)	2 (5.9)	5 (15.6)	7 (7.1)	
≥4	100 (32.2)	21 (14.3)	19 (55.9)	1 (3.1)	59 (60.2)	
Location of metastasis **						
Brain	94 (30.2)	34 (23.1)	12 (35.3)	8 (25.0)	40 (40.8)	0.023 ^c^
Bone	139 (44.7)	67 (45.6)	12 (35.3)	10 (31.3)	50 (51.0)	0.159
Intrathoracic metastasis ^†^	192 (61.7)	79 (53.7)	22 (64.7)	23 (71.9)	68 (69.4)	0.050
Intraabdominal metastasis ^‡^	74 (23.8)	38 (25.9)	6 (17.6)	3 (9.4)	27 (27.6)	0.140
Others ^§^	14 (4.5)	8 (5.5)	0 (0.0)	0 (0.0)	6 (6.1)	0.337
Tissue EGFR mutation type ^∥^						
Exon 19 deletion	68 (21.9)	0 (0.0)	0 (0.0)	13 (40.6)	55 (56.1)	<0.001 ^bcde^
L858R (exon 21)	54 (17.4)	0 (0.0)	0 (0.0)	12 (37.5)	42 (42.9)	<0.001 ^bcde^
Others ^¶^	17 (5.5)	0 (0.0)	0 (0.0)	6 (18.8)	11 (11.2)	<0.001 ^bc^
Plasma EGFR mutation type ^∥^						
Exon 19 deletion	71 (22.8)	0 (0.0)	17 (50.0)	0 (0.0)	54 (55.1)	<0.001 ^acdf^
L858R (exon 21)	58 (18.6)	0 (0.0)	16 (47.1)	0 (0.0)	42 (42.9)	<0.001 ^acdf^
Others ^++^	6 (1.9)	0 (0.0)	1 (2.9)	0 (0.0)	5 (5.1)	0.019 ^c^
Use of EGFR-TKI	154 (49.5)	0 (0.0)	32 (94.1)	26 (81.3)	96 (98.0)	<0.001 ^abcf^
Gefitinib	46 (14.8)	0 (0.0)	11 (32.3)	12 (37.5)	23 (23.5)	<0.001 ^abc^
Erlotinib	23 (7.4)	0 (0.0)	4 (11.8)	2 (6.3)	17 (17.3)	<0.001 ^ac^
Afatinib	77 (24.7)	0 (0.0)	14 (41.2)	10 (31.2)	53 (54.1)	<0.001 ^abc^
Osimertinib	8 (2.6)	0 (0.0)	3 (8.8)	2 (6.3)	3 (3.1)	0.014 ^a^
Use of EGFR-TKI as 1st line treatment	147 (47.3)	0 (0.0)	29 (85.3)	25 (78.1)	93 (94.9)	<0.001 ^abc^

Data are presented as n (%) or as median (interquartile range). EGFR, epidermal growth factor receptor; DNA, deoxyribonucleic acid; BMI, body mass index; ECOG, European Cooperative Oncology Group; CEA, carcinoembryonic antigen; NCSCLC-NOS, non-small cell lung cancer–not otherwise specified; TKI, tyrosine kinase inhibitor. * Eight patients without tissue biopsy were confirmed by histology or cytology of pleural effusion (n = 6), pericardial effusion (n = 1), or bronchial washing fluid (n = 1). ** Patients could have more than one metastatic site. ^†^ Including lung-to-lung, pleura, and pericardium metastasis. ^‡^ Including liver, abdominal lymph node, adrenal gland, kidney, and peritoneum metastasis. ^§^ Including axillary lymph node, thyroid, and muscle metastasis. ^∥^ Some patients could have two mutation types. ^¶^ Insertion mutation in exon 20 (n = 10), T790M (exon 20) (n = 2), G719X (exon 18) (n = 2), S768I (exon 20) & L861Q (exon 21) (n = 1), S768I (exon 20) (n = 1), or L861Q (exon 21) (n = 1); ^++^ Insertion mutation in exon 20 (n = 1), T790M (exon 20) (n = 1), S768I (exon 20) (n = 2), or L861Q (exon 21) (n = 2); ^a^
*p* < 0.05 with Bonferroni correction between “Tissue EGFR (−) & plasma EGFR (−)” and “Tissue EGFR (−) & plasma EGFR (+)”. ^b^
*p* < 0.05 with Bonferroni correction between “Tissue EGFR (−) & plasma EGFR (−)” and “Tissue EGFR (+) & plasma EGFR (−)”. ^c^
*p* < 0.05 with Bonferroni correction between “Tissue EGFR (−) & plasma EGFR (−)” and “Tissue EGFR (+) & plasma EGFR (+)”. ^d^
*p* < 0.05 with Bonferroni correction between “Tissue EGFR (−) & plasma EGFR (+)” and “Tissue EGFR (+) & plasma EGFR (−)”. ^e^
*p* < 0.05 with Bonferroni correction between “Tissue EGFR (−) & plasma EGFR (+)” and “Tissue EGFR (+) & plasma EGFR (+)”. ^f^
*p* < 0.05 with Bonferroni correction between “Tissue EGFR (+) & plasma EGFR (−)” and “Tissue EGFR (+) & plasma EGFR (+)”.

**Table 2 jcm-11-01144-t002:** Factors associated with a positive result of plasma EGFR mutation analysis (N = 311).

Variables	Univariable		Multivariable	
	Unadjusted OR (95% CI)	*p*-Value	Adjusted OR (95% CI)	*p*-Value
Age, years	0.99 (0.97–1.01)	0.242	0.98 (0.95–1.00)	0.086
Sex, female	1.85 (1.16–2.94)	0.009		
Smoking history				
Ever smoker	Reference		Reference	
Never smoker	2.59 (1.59–4.24)	< 0.001	2.83 (1.55–5.20)	0.001
CEA level				
<3.2 ng/mL	Reference		Reference	
3.2–94.7 ng/mL	3.22 (1.55–6.68)	0.002	2.61 (1.16–5.84)	0.020
>94.7 ng/mL	3.89 (1.75–8.62)	0.001	2.98 (1.21–7.35)	0.018
N stage				
N0	Reference		Reference	
N1	1.50 (0.43–5.24)	0.525	1.65 (0.38–7.17)	0.501
N2	2.29 (0.88–5.91)	0.088	2.52 (0.78–8.17)	0.124
N3	4.23 (1.76–10.20)	0.001	4.22 (1.41–12.62)	0.010
M stage				
M1a	Reference			
M1b	1.07 (0.45–2.57)	0.877		
M1c	2.23 (1.31–3.78)	0.003		
Type of metastatic organs *				
Brain	2.12 (1.30–3.47)	0.003	2.73 (1.39–5.36)	0.003
Bone	1.14 (0.72–1.79)	0.575		
Intrathoracic metastasis **	1.62 (1.01–2.59)	0.045	2.61 (1.38–4.96)	0.003
Intraabdominal metastasis ^†^	1.12 (0.66–1.90)	0.668	1.85 (0.93–3.68)	0.079
Others ^‡^	1.01 (0.34–2.99)	0.983		

DNA, deoxyribonucleic acid; EGFR, epidermal growth factor receptor; OR, odds ratio; CI, confidence interval; BMI, body mass index; CEA, carcinoembryonic antigen. * Patients could have more than one metastatic site. ** Including lung to lung, pleura, and pericardium metastasis. ^†^ Including liver, abdominal lymph node, adrenal gland, kidney, and peritoneum metastasis. ^‡^ Including axillary lymph node, thyroid, and muscle metastasis.

**Table 3 jcm-11-01144-t003:** Details of diagnostic procedures (N = 311).

Variables	N = 311
Tissue biopsy	
Yes	303 (97.4)
No	8 (2.6)
Biopsy methods	
EBUS-TBNA	146 (46.9)
Percutaneous core needle biopsy	99 (31.8)
TBLB	38 (12.2)
VATS	16 (5.1)
Others *	4 (1.3)
Biopsy sites	
Mediastinal lymph nodes	146 (46.9)
Lung	103 (33.1)
SCN	33 (10.6)
Others ^†^	21 (6.8)
Tissue EGFR mutation test	
Performed	251 (80.7)
Time taken from the first hospital visit to the test, days	7 (4–12)
EGFR gene, mutation real-time PCR	236 (75.9)
Turnaround time, days	11 (9–13)
NGS	15 (4.8)
Turnaround time, days	41 (34–49)
Not performed	60 (19.3)
Result of tissue EGFR mutation test (N = 251)	
Positive	130 (51.8)
Negative	121 (48.2)
Plasma EGFR mutation	
Performed	311 (100)
Time taken from the first hospital visit to the test, days	1 (0–5)
Turnaround time, days	5 (4–6)
Result of plasma EGFR mutation test (N = 311)	
Positive	132 (42.4)
Negative	179 (57.6)

Data are presented as n (%) or median (interquartile range). EBUS-TBNA, endobronchial ultrasound guided-transbronchial needle aspiration; TBLB, transbronchial lung biopsy; VATS, video-assisted thoracic surgery; SCN, supraclavicular lymph node; EGFR, epidermal growth factor receptor; PCR, polymerase chain reaction. * Metastatic organ surgical biopsy: pericardium (n = 2), brain (n = 1), bone (n =1). ^†^ Bone (n = 4), pleura (n = 7), liver (n = 3), axillary lymph node (n = 2), mediastinal and chest wall mass (n = 2), pericardium (n = 2), brain (n = 1).

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
