# Peer review of "Clinical Utility of Plasma Cell-Free DNA EGFR Mutation Analysis in Treatment-Naïve Stage IV Non-Small Cell Lung Cancer Patients"

_jcm, 2022, doi:10.3390/jcm11041144_

Round 1
Reviewer 1 Report
In this manuscript Kim and colleagues report the clinical use of plasma cell-free DNA EGFR mutation analysis in treatment-naive stage IV NSCLC patients. They demonstrate that plasma EGFR mutation was significantly related to never-smoking status, N3 stage, carcinoembryonic antigen levels and brain or intrathoracic metastasis.
The manuscript is interesting and novel. I agree with the authors that this study has some limitations. In the future, it is important to assess whether the plasma EGFR mutation and the associated factors are reflecting to a more generic population.
Author Response
Thank you so much for your positive feedback.
Reviewer 2 Report
Authors examined the clinical utility of plasma cell-free DNA EGFR mutation analysis using non-small cell lung cancer patients. Current cohort study revealed that smoking status and the factors reflecting tumor burden were associated with detection of plasma EGFR mutation, which further can shorten turnaround time and have an advantage with high 2-year overall survival ratio. This finding would accelerate the clinical use of plasma cell-free DNA EGFR, while more scientific insights should be included in this manuscript. For example, application of any other plasma cell-free DNAs to other types of tumors, backgrounds about the relationship between EGFR mutation and metastasis, and subgrouping validities (this may partly be state in Discussion section showing other report). In total, this manuscript would have informative things for proceeding application of diagnosis using plasma cell-free DNA to NSCLC, even though above-mentioned minor issues should be referred.
Author Response
R: Thank you so much for your valuable comments. As the reviewer recommended to include more scientific insights, we have added them in the revised manuscript.
R1: We have added following sentences for application of any other plasma cell-free DNAs to other types of tumors in the Discussion Section as follows;
“Clinical application of liquid biopsy using plasma cfDNA have increased rapidly for various solid tumors such as lung cancer, breast cancer, colon cancer, melanoma, stomach, etc [1,2]. Plasma cfDNA technologies have the potential to identify actionable somatic alterations, tumor mutational burden, mutational signature and tumor-associated methylation changes [1]. Liquid biopsy using plasma cfDNA can influence on the early detection, monitoring of minimal residual disease, evaluation of early treatment response, and assessment of clonal evolution of various solid tumors [3].”
R2: We have added following sentences for the relationship between EGFR mutation and metastasis in the Discussion Section as follows;
“Previous studies have demonstrated that the presence of EGFR mutations is associated with distant metastases [4,5]. In addition, previous studies also reported that the presence of EGFR mutation is associated with bone, brain, and liver metastases [4-8].”
R3: For subgrouping validities, we already addressed the relationship between the patient characteristics and the detection of the plasma EGFR mutation in the Discussion section. However, we have revised the manuscript to provide more scientific insights as follows;
Before: “Previous studies have evaluated the relationship between patient characteristics and the detection of the plasma EGFR mutation. These studies suggested that lymph node involvement, extrathoracic metastasis, and metastasis to bone or liver were associated with the detection of plasma EGFR mutation. In our study, the detection of plasma EGFR mutation was also associated with a higher tumor burden such as a higher N stage, and brain or intrathoracic metastasis.”
After: “Previous studies have evaluated the relationship between patient characteristics and the detection of the plasma EGFR mutation [9-13]. One French multicenter study reported that the detection rate of EGFR mutation in plasma test was about 44 times higher in never-smoker than in ever-smoker [9]. Previous studies also revealed that the detection of plasma EGFR mutation was approximately 3 - 9 times higher in the patients with higher N stage [10, 13], 5 times higher in the patients with extrathoracic metastasis [10], and 4-10 times higher in the patients with distant organ metastasis such as bone [12, 13]. In our study, similar to previous studies, the detection of plasma EGFR mutation was associated with never-smoker and higher tumor burden such as a higher N stage, and brain or intrathoracic metastasis.”
References
- Cescon DW, Bratman SV, Chan SM, Siu LL. Circulating tumor DNA and liquid biopsy in oncology. Nat Cancer. 2020;1(3):276-290.
- Palmirotta R, Lovero D, Cafforio P, Felici C, Mannavola F, Pellè E, Quaresmini D, Tucci M, Silvestris F. Liquid biopsy of cancer: a multimodal diagnostic tool in clinical oncology. Ther Adv Med Oncol. 2018;10:1758835918794630.
- Siravegna G, Mussolin B, Venesio T, Marsoni S, Seoane J, Dive C, Papadopoulos N, Kopetz S, Corcoran RB, Siu LL, Bardelli A. How liquid biopsies can change clinical practice in oncology. Ann Oncol. 2019;30(10):1580-1590.
- Kim, T.; Kim, E.Y.; Lee, S.H.; Kwon, D.S.; Kim, A.; Chang, Y.S. Presence of mEGFR ctDNA predicts a poor clinical outcome in lung adenocarcinoma. Thorac Cancer 2019, 10, 2267-2273.
- Sholl, L.M.; Aisner, D.L.; Varella-Garcia, M.; Berry, L.D.; Dias-Santagata, D.; Wistuba, II; Chen, H.; Fujimoto, J.; Kugler, K.; Franklin, W.A.; et al. Multi-institutional Oncogenic Driver Mutation Analysis in Lung Adenocarcinoma: The Lung Cancer Mutation Consortium Experience. J Thorac Oncol 2015, 10, 768-777.
- Guan, J.; Chen, M.; Xiao, N.; Li, L.; Zhang, Y.; Li, Q.; Yang, M.; Liu, L.; Chen, L. EGFR mutations are associated with higher incidence of distant metastases and smaller tumor size in patients with non-small-cell lung cancer based on PET/CT scan. Med Oncol 2016, 33, 1.
- Yang, B.; Lee, H.; Um, S.W.; Kim, K.; Zo, J.I.; Shim, Y.M.; Jung Kwon, O.; Lee, K.S.; Ahn, M.J.; Kim, H. Incidence of brain metastasis in lung adenocarcinoma at initial diagnosis on the basis of stage and genetic alterations. Lung Cancer 2019, 129, 28-34.
- Doebele, R.C.; Lu, X.; Sumey, C.; Maxson, D.A.; Weickhardt, A.J.; Oton, A.B.; Bunn, P.A., Jr.; Barón, A.E.; Franklin, W.A.; Aisner, D.L.; et al. Oncogene status predicts patterns of metastatic spread in treatment-naive nonsmall cell lung cancer. Cancer 2012, 118, 4502-4511.
- Denis, M.G.; Lafourcade, M.P.; Le Garff, G.; Dayen, C.; Falchero, L.; Thomas, P.; Locher, C.; Oliviero, G.; Licour, M.; Reck, M.; et al. Circulating free tumor-derived DNA to detect EGFR mutations in patients with advanced NSCLC: French subset analysis of the ASSESS study. J Thorac Dis 2019, 11, 1370-1378.
- Seki, Y.; Fujiwara, Y.; Kohno, T.; Yoshida, K.; Goto, Y.; Horinouchi, H.; Kanda, S.; Nokihara, H.; Yamamoto, N.; Kuwano, K.; et al. Circulating cell-free plasma tumour DNA shows a higher incidence of EGFR mutations in patients with extrathoracic disease progression. ESMO Open 2018, 3, e000292.
- Ikushima, H.; Sakatani, T.; Usui, K. Clinical Features of Patients with an Epidermal Growth Factor Receptor T790M Mutation Detected in Circulating Tumor DNA. Oncology 2020, 98, 23-28.
- Lee, Y.; Park, S.; Kim, W.S.; Lee, J.C.; Jang, S.J.; Choi, J.; Choi, C.M. Correlation between progression-free survival, tumor burden, and circulating tumor DNA in the initial diagnosis of advanced-stage EGFR-mutated non-small cell lung cancer. Thorac Cancer 2018, 9, 1104-1110.
- Kuo, C.Y.; Lee, M.H.; Tsai, M.J.; Yang, C.J.; Hung, J.Y.; Chong, I.W. The Factors Predicting Concordant Epidermal Growth Factor Receptor (EGFR) Mutation Detected in Liquid/Tissue Biopsy and the Related Clinical Outcomes in Patients of Advanced Lung Adenocarcinoma with EGFR Mutations. J Clin Med 2019, 8.
